# Cold Stress Induced Liver Injury of Mice through Activated NLRP3/Caspase-1/GSDMD Pyroptosis Signaling Pathway

**DOI:** 10.3390/biom12070927

**Published:** 2022-07-01

**Authors:** Yuanyuan Liu, Nianyu Xue, Boxi Zhang, Hongming Lv, Shize Li

**Affiliations:** National Experimental Teaching Demonstration Center of Animal Medicine Foundation, College of Animal Science and Veterinary Medicine, Heilongjiang Bayi Agricultural University, Daqing 163319, China; lyy3648@163.com (Y.L.); xuenianyu1994@163.com (N.X.); zhangboxi0304@163.com (B.Z.)

**Keywords:** cold stress, pyroptosis, inflammasome, apoptosis, liver injury

## Abstract

The body needs to generate heat to ensure basic life activities when exposed to cold temperatures. The liver, as the largest glycogen storage organ in the body and main heat-producing organ at rest, may play a role in chronic cold exposure. Recent studies suggested that pyroptosis plays a crucial role in liver diseases. However, the role of pyroptosis in cold stress-induced liver injury is not clear. Hence, in this study, we attempted to investigate the effects of chronic cold exposure on liver function, apoptosis, oxidative stress and inflammation in mice by establishing a mouse model of chronic cold exposure, and to investigate whether pyroptosis pathways are involved in the process of chronic cold exposure. In vivo, our results show that inflammatory cell infiltration and other pathological changes in liver cells and the activity of liver enzyme evidently increased in the serum and liver of cold-exposed mice, suggesting cold stress may result in liver injury. Remarkably, increased expression of heat shock protein 70 (HSP70) and HSP90 proteins proved the cold stress model is successfully constructed. Then, elevated levels of apoptosis, inflammation, oxidative stress and pyroptosis related proteins and mRNAs, such as cysteinyl aspartate specific proteinase-3 (Caspase-3), inducible nitric oxide synthase (iNOS), nuclear factor erythroid2-related factor 2 (Nrf2) and gasdermins D (GSDMD), confirmed that cold exposure activated apoptosis, oxidative stress and pyroptosis, and released inflammation cytokines. Meanwhile, in vitro, we got similar results as in vivo. Further, adding an NLR family pyrin domain containing 3 (NLRP3) inhibitors found that suppression expression of NLRP3 results in the essential proteins of pyroptosis and antioxidant evidently reduced, and adding GSDMD inhibitor found that suppression expression of GSDMD accompanies with the level of Nrf2 and heme oxygenase-1 (HO-1) obviously reduced. In summary, these findings provide a new understanding of the underlying mechanisms of the cold stress response, which can inform the development of new strategies to combat the effects of hypothermia.

## 1. Introduction

Numerous studies have shown that chronic stress can lead to an imbalance in the homeostasis of the physiological internal environment and induce a variety of primary diseases, such as cancer and neurological disorders [1,2]. For example, chronic cold stress is becoming a growing problem for humans and animals that must work or stay in extreme environments, leading to disruptions in energy metabolism and dysfunction of the immune and endocrine systems [3,4]. Cold stress has been reported to be closely associated with several diseases; for example, chronic cold stress at 4 °C induces cardiac remodeling and dysfunction via the endothelin system [5]; intermittent cold stress induces dysfunction of the sympathetic adrenal medulla in rats [6]. Moreover, cold stress is one of the important causes of serious damage to the economic benefit of livestock and poultry breeding [7]. For instance, Jianyong Wang et al. found that when the feeding temperature decreased, the feed intake of fattening pigs obviously increased, but the bodyweight decreased, or slow growth occurred instead [8]. In our previous study, cold stress in mice induced liver injury and oxidative stress [9]. However, it is still not clear what mechanisms induced liver injury.

It is well known that pyroptosis is a form of cell death accompanied by an inflammatory response and is a new way of programmed cell death induced by the inflammasome [10,11]. Pyroptosis is characterized by rapid rupture of the plasma membrane, DNA damage and the release of intracellular pro-inflammatory substances [12]. The activated pyroptosis signaling pathway can be mainly divided into the Caspase-1-dependent classical pathway and the non-Caspase-1-dependent non-classical pathway [13]. When the classical pyroptosis signaling pathway is activated, cells swell and activated caspase-1 cleaves the full-length GSDMD and releases the GSDMD-N terminus, terminating its inhibitory effect, which then oligomerizes and assembles into pores on the plasma membrane, leading to a massive release of cellular contents and induction of inflammatory responses [14]. Increasing evidence suggests that the NLRP3 inflammatory activation mediated Caspase-1-dependent classical pathway is an essential driver of various acute and chronic liver diseases [15]. Alexander Wree et al. discovered that over-expression of NLRP3 can induce Caspase-1-dependent mouse hepatocyte pyroptosis, inflammation and fibrosis [16]. Furthermore, Bing Xu and his colleague proved that increased expression of GSDMD is related to lobular inflammation and ballooning in the human non-alcoholic steatohepatitis (NASH) liver [17]. Moreover, activated oxidant stress is mainly characteristic of cold stress. Oxidation of unsaturated fatty acids, and increased non-enzymatic antioxidants and lipid peroxidation in different tissues (such as liver, kidney and brain tissues), in turn cause membrane damage, often disrupting tissue integrity [18]. Reactive oxygen species (ROS) activation is a major pathway for NLRP3/Caspase-1-dependent activation of the pyroptosis signaling pathway [19]. Alcohol, viruses or other toxic stimuli can damage hepatocytes, leading to the release of excess ROS for oxidative stress, followed by activation of inflammatory vesicles via the thioredoxin-interacting protein (TXNIP) signaling pathway and further through Caspase-1 and Caspase-11, respectively, to induce pyroptosis [20]. However, the relationship between oxidative stress and pyroptosis involved in the development of cold stress is still not clear.

Whether the process of pyroptosis occurs in the mouse liver following cold exposure is not known. In this study, we investigated the phenomenon of liver injury, the levels of inflammation, oxidative stress, apoptosis and pyroptosis-related proteins and mRNAs expression in the mouse and AML12 cells after cold exposure. We also characterized the relationship between oxidative stress and the pyroptosis signaling pathway in liver AML12 cells following cold exposure by adding an NLRP3 inhibitor and GSDMD inhibitor. We provided primary data to clarify the mechanisms by which cold stress might affect liver injury.

## 2. Materials and Methods

### 2.1. Animals and Ethics

The male C57BL/6 mice (3 weeks old) were purchased from Changchun National Laboratory Animal Center (China). Mice were fed for a week to acclimatize before the experiment began. During feeding, the mice could eat and drink freely and follow the cycle of day and night, providing 12 h of light and 12 h of darkness (turned on the lights at 8:00 a.m. and turn off at 8:00 p.m. at regular intervals every day), and the lighting intensity was 200 Lux. Raised the temperature of 24 ± 0.5 °C with 40 ± 5% relative humidity. All animal testing procedures were approved by the Academic Committee of Heilongjiang Bayi Agricultural University (Daqing, China).

### 2.2. Experimental Design and Sample Collection

Mice were randomly divided into two groups: the room temperature (Control) group (*n* = 6) and the cold exposure (Cold) group (*n* = 6). The mice of the cold group were exposed to cold ambient temperature at 4 °C random 3 h a day for four consecutive weeks, while the control group was at 24 ± 0.5 °C. The specific experimental design is shown in Figure 1. After cold exposure, we collected blood and liver samples and put the blood sample in a refrigerator at 4 °C. The part of liver sample was frozen in liquid nitrogen and stored at −80 °C for subsequent examination, the rest of liver sample was fixed in 4% paraformaldehyde (Beyotime, Shanghai, China).

### 2.3. Enzyme-Linked Immunosorbent Assay (ELISA)

Liver tissues were ground thoroughly and collected the supernatant of the homogenate centrifuged. Then used BCA protein assay (Beyotime, Shanghai, China) to determine the protein content in the supernatant and serum. We used ELISA kit (Nanjing JianCheng Bioengineering Institute, Nanjing, China) to detect the alanine aminotransferase (ALT) activity, aspartate aminotransferase (AST) activity, lactate dehydrogenase (LDH) activity, catalase from micrococcus lysodeiktic (CAT) activity, glutathione (GSH) content, malondialdehyde (MDA) content.

### 2.4. Hematoxylin-Eosin (HE) Staining

The liver tissue was fixed in 4% paraformaldehyde, dehydrated with gradient ethanol and cleared in xylene, then embedded in paraffin and cut into slices of approximately 3–4 μM with a microtome. The sections were then subjected to dewaxing in xylene, rehydration in gradient ethanol, and dying with hematoxylin in 15 min. Next, the sections were differentiated in 1% hydrochloric acid/ethanol, re-stained in 0.5% ammonia, and dyed with eosin. After that, the sections were dehydrated with gradient ethanol, cleared with xylene, and blocked in a neutral balsam, followed by observing under an optical microscope.

### 2.5. Immunohistochemistry

Liver tissues were paraffin-embedded, sectioned (3–4 µM) were rinsed with PBS three times. The sections were blocked with 0.3% H_2_O_2_ for 15 min and the endogenous peroxidase was inactivated, and then rinsed with PBS three times. The next step was to block sections with 5% bovine serum albumin (BSA) for 20 min at room temperature and incubate them with the primary antibody, including NLRP3 (1:200, A14223, ABclonal, Wuhan, China), GSDMD (1:200, A18281, ABclonal, Wuhan, China), Caspase-1 (1:500, A16792, ABclonal, Wuhan, China), Nrf2 (1:200, A0674, ABclonal, Wuhan, China), at 4 °C overnight. The following day, HRP-labeled Goat Anti-Rabbit IgG (H + L) (1:1000, SA00001-2, Proteintech, Chicago, IL, USA) was added and incubated at room temperature for 60 min. Then, the sections were washed by PBS three times, and dehydrated by alcohol gradient, cleared in xylene, and put under the microscope to detect and take photos.

### 2.6. Western Blot Analysis

The 20 µg of total proteins were isolated from AML12 cells, and mouse livers with NP40 lysis buffer (Beyotime, Shanghai, China) (with 1 mM PMSF). The protein concentration of the supernatant was determined using a BCA reagent (Beyotime, Shanghai, China). Then used 10–12.5% sodium dodecyl sulfate-polyacrylamide (SDS-PAGE) and subsequently transferred onto polyvinylidene fluoride (PVDF) membranes (0.22 and 0.45 µm, Millipore, Darmstadt, Germany) for antibody blotting. The primary antibodies used were against NLRP3 (1:1000, A14223, ABclonal, Wuhan, China), toll-like receptors 4 (TLR4) (1:1000, A5258, ABclonal, Wuhan, China), GSDMD (1:1000, A18281, ABclonal, Wuhan, China), Caspase-1 (1:1000, A16792, ABclonal, Wuhan, China), interleukin-1β (IL-1β) (1:1000, A11369, ABclonal, Wuhan, China), IL-18 (1:1000, A16737, ABclonal, Wuhan, China), Nrf2 (1:1000, A0674, ABclonal, Wuhan, China), apoptosis-associated speck-like protein containing CARD (ASC) (1:1000, A16672, ABclonal, Wuhan, China), iNOS (1:1000, A0312, ABclonal, Wuhan, China), cyclooxygenase-2 (COX-2) (1:1000, A14223, ABclonal, Wuhan, China), high mobility group protein (HMGB1) (1:1000, A2553, ABclonal, Wuhan, China); HO-1 (1:1000, #ab68477, Abcam, Cambridge, London, UK), NAD(P)H quinone dehydrogenase 1 (NQO1) (1:1000, #ab80588, Abcam, London, UK), glutamate cysteine ligase (GCLC) (1:1000, #ab207777, Abcam, London, UK), GCLM (1:1000, #ab126704, Abcam, London, UK), Bcl-2-associated X (Bax) (1:1000, #ab32503, Abcam, London, UK), B-cell lymphoma-2 (Bcl-2) (1:1000, #ab59348, Abcam, London, UK), Tubulin (1:1000, #ab6046, Abcam, London, UK); tumor necrosis factor-α (TNF-α) (1:1000, 346654, ZEN BIO, Chengdu, China), HSP70 (1:1000, R24632, ZEN BIO, Chengdu, China); HSP90 (1:1000, 60318-1-lg, Proteintech, Chicago, IL, USA), Caspase-3 (1:1000, 300968, ZEN BIO, Chengdu, China), β-actin (1:1000, 60008-1-Ig, Proteintech, Chicago, IL, USA), HRP-labeled Goat Anti-Mouse IgG(H + L) (1:1000, SA00001-1, Proteintech, Chicago, IL, USA), HRP-labeled Goat Anti-Rabbit IgG(H + L) (1:1000, SA00001-2, Proteintech, Chicago, USA). Membranes were blocked using 5% nonfat milk for 1 h and incubated with primary antibodies overnight at 4 °C. The next day, membranes were washed thrice with Tris-buffered saline with 0.1% Tween 20 (TBST), then incubated with secondary antibodies (Proteintech, Chicago, USA) for 1 h at room temperature. Target proteins were visualized with an enhanced chemiluminescence kit (ECL, MILIBO, Colorado Springs, CO, MA, USA), and each band was analyzed by ImageJ software (Bio-Rad Laboratories, Hercules, CA, USA).

### 2.7. Quantitative RT-PCR (qRT-PCR)

Total RNA was extracted from liver and AML12 cells using Trizol reagent (TAKARA, Maebashi, Kyoto, Japan). Then the RNA was reverse transcribed to cDNA using a PrimeScript RT Reagent Kit (TAKARA, Kyoto, Japan). A qRT-PCR was performed with the SYBR Premix Ex Taq™ (TAKARA, Kyoto, Japan). We used β-actin as the internal control. In addition, relative RNA was calculated using 2^−ΔΔCt^ normalized to β-actin. The sequences of qRT-PCR primers for the genes examined are listed in Table A1.

### 2.8. Cell Culture and Reagents

Mouse hepatocyte’s AML12 cells were purchased from Shanghai Cell Bank, Chinese Academy of Sciences. The cells were maintained in DMEM high sugar liquid medium (C11995500BT, Thermo Fisher Scientific, Waltham, MA, USA) with 10 % Fetal Bovine Serum (FBS) (C0232, Thermo Fisher Scientific, Waltham, MA, USA), 100 U/mL penicillin and 100 μg/mL streptomycin (Beyotime, Shanghai, China), and incubation at 37 °C in a 5% CO_2_ humidified incubator. Change the cell fluid every day and passaged two times a week.

### 2.9. Cell Viability Assay

AML12 cells were seeded in 96-well culture plates at a density of 5 × 10^3^ cells/well and divided into six groups: the 37 °C (Control 12 h; Control 24 h; Control 36 h) groups; the mild hypothermia (32 °C) (MH 12 h; MH 24 h; MH 36 h) groups and cultured 24 h. Subsequently, 10% 3-(4,5-dimethyl-2-thiazolyl)-2,5-diphenyl-2-H-tetrazolium bromide (MTT) solution (5 mg/mL) (M2128, Sigma, Darmstadt, Germany) was added to each well. After 3 h, the absorbance was measured at 570 nm in a microplate reader to determine the cell viability.

### 2.10. Cell Morphology

AML12 cells were seeded in 36 mm culture plates and cultured 24 h at a density of 1 × 10^6^ cells/well and divided into six groups: the 37 °C (Control 12 h; Control 24 h; Control 36 h) groups; the mild hypothermia (32 °C) (MH 12 h; MH 24 h; MH 36 h) groups. After then, the AML12 cells were observed by an inverted microscope and took photos.

### 2.11. Hoechst 33258 Staining

AML12 cells were seeded in 36 mm culture plates and cultured 24 h at a density of 2.5 × 10^5^ cells/well and divided into six groups: the 37 °C (Control 12 h; Control 24 h; Control 36 h) groups; the mild hypothermia (32 °C) (MH 12 h; MH 24 h; MH 36 h) groups. After then, the AML12 cells were incubated in the dark at 37 °C with 5 mL of 0.5 µg/mL Hoechst 33,258 staining solution for 15 min. Then, the cells were washed by PBS three times. Finally, the AML12 cells were observed by an inverted fluorescence microscope and took photos.

### 2.12. ROS Assay

AML12 cells were seeded in 96-well culture plates and cultured 24 h at a density of 1 × 10^3^ cells/well and divided into six groups: the 37 °C (Control 12 h; Control 24 h; Control 36 h) groups; the mild hypothermia (32 °C) (MH 12 h; MH 24 h; MH 36 h) groups. After then, the AML12 cells were incubated in the dark at 37 °C with 100 µL of DCFDA staining solution for 30 min. Then, the cells were washed by PBS three times. Finally, the AML12 cells were measured by fluorescence microplate (excitation light is 504 nm; emitted light is 529 nm) and observed by an inverted fluorescence microscope and took photos.

### 2.13. Add Inhibitor

AML12 cells were seeded in 36 mm culture plates and cultured 24 h at a density of 1 × 10^6^ cells/well and divided into four groups: the 37 °C (Control; Control + inhibitor) groups; the mild hypothermia (32 °C) (MH; MH + inhibitor) groups. After adding inhibitor 3 h, put the MH and MH + inhibitor groups of AML12 cells in a 32 °C incubator for 6 h. Finally, collected cells for subsequent experiments. The inhibitors are NLRP3 inhibitor (MCC950, final concentration is 100 µM) and GSDMD inhibitor (Necrosulfonamide, final concentration is 20 µM).

### 2.14. Statistical Analysis

All data were analyzed by the GraphPad Prism 6.0 software (San Diego, CA, USA) and expressed by mean ± standard deviation (SD). Statistical comparisons were assessed across different treatment groups (room temperature and cold exposure). All data analyses were performed using one-way analysis of variance (ANOVA). *p* < 0.05 was considered statistically significant.

## 3. Results

### 3.1. The Effect of Cold Exposure on the Liver Function in the Mice Liver

To investigate the impact on the liver following cold exposure, we stained the mice liver by HE (Figure 2A). In the control group, the hepatocytes were well arranged, with normal morphology, prominent nuclei and uniform cytoplasmic staining. However, more hepatocytes appeared to degenerate in the cold group, swollen or even necrotic and disappeared, with no conspicuous intercellular demarcation (Figure 2A). The liver enzyme activity was distinctly increased after cold exposure in the serum and liver, including ALT, AST and LDH (Figure 2B,C,E–G), except for the activity of LDH in the serum (Figure 2D). Figure 1 shows the associations between cold stress and heat shock proteins (HSPs). Cold stress-induced HSP70 and HSP90 expression in the liver were tested by Western blot (Figure 2H–J). In addition, for cold stress-induced apoptosis, as shown in Figure 1, the expression ratios of Bax/Bcl-2 and Cleaved-Caspase-3/Pro-Caspase-3 were remarkably increased (Figure 2K–M).

### 3.2. The Effect of Cold Exposure on the Expression of Inflammatory Cytokines and Oxidative Stress-Related Indexes in the Mice Liver

In order to assay the effects of cold exposure on inflammatory related proteins in the liver of mice, we measured the expression of TLR4, iNOS, COX-2, TNF-α and HMGB1 by Western blot and qRT-PCR. Figure 3 shows that TLR4, iNOS, COX-2, TNF-α and HMGB1 protein levels were up-regulated in the cold groups relative control groups (Figure 3A–F). In addition, mRNAs expression data demonstrated that TLR4, TNF-α and HMGB1 gene expression were activated after cold exposure compared with control groups (Figure 3G–I). It is well-known that CAT, GSH and MDA were the markers of oxidation stress that can reflect the level of oxidation. Here, ELISA results demonstrated that CAT, GSH activity was apparently decreased and MDA content was distinctly increased in the serum of cold groups compared with control groups (Figure 3J–L). In addition, in the mice liver of cold groups, the GSH activity was also obviously decreased, and MDA content was also outstanding increased (Figure 3N,O). However, the activity of CAT had no significant differences (Figure 3M). Then, the proteins of the anti-oxidative stress relevant signaling pathways were also measured in the liver. The results showed an increase in Nrf2, HO-1, NQO1, GCLC and GCLM proteins expression of cold groups compared with control groups (Figure 3Q–U). In addition, the mRNA expression also increased in Nrf2, HO-1, NQO1, GCLC and GCLM (Figure 3V–Z). Immunohistochemistry results revealed the content of Nrf2 in the liver (Figure 3A), and the cold group had more levels of Nrf2 than the control group (Figure 3B).

### 3.3. The Effect of Cold Exposure on the NLRP3/GSDMD Pyroptosis Pathway in the Mice Liver

As shown in Figure 4, cold stress resulted in a significant increase in the NLRP3/GSDMD pyroptosis pathway. The Western blot result shows that NLRP3, ASC, Caspase-1, GSDMD, IL-1β, IL-18 protein levels were up-regulated in the cold groups relative control groups (Figure 4A–G). In addition, mRNA expression data demonstrated that NLRP3, Caspase-1, GSDMD, IL-1β and IL-18 gene expression were activated after cold exposure compared with control groups (Figure 4H–L). Furthermore, immunohistochemistry results further revealed more content of NLRP3, Caspase-1, GSDMD in the liver (Figure 4M–P).

### 3.4. The Effect of Hepatocytes in the AML12 Cells following Mild Hypothermia Treatment

To further confirm the influence of MH on mouse hepatocytes, the morphology of AML12 cells was observed under a microscope at first. Figure 5 shows a marked change after MH treated 36 h, including a significant reduction in the number of cells, more cellular debris at the bottom of the medium and “drawing” of the cells morphologically (Figure 5A). In addition, the MTT assay result shows that the activity of AML12 cells was dropped clearly following MH treatment at different times, and there is a particular time dependency (Figure 5B). The results were consistent with in vivo experiments; MH treatment for 6 h resulted in a HSP70 protein level distinctly increased, and treatment for 3 h resulted in HSP90 protein expression that was visibly increased (Figure 5C–E). Hoechst 33258 staining was performed to identify the influence of MH treatment on AML12 cell apoptosis. The results showed that AML12 cells had fragmentary dense staining after MH treatment, indicating that MH led to cell apoptosis (Figure 5F). In addition, Western blot results also proved that MH treatment activated apoptosis-related protein expression in the AML12 cells, including expression ratios of Bax/Bcl-2 and Cleaved-Caspase-3/Pro-Caspase-3 (Figure 5G–I).

### 3.5. The Effect of Inflammatory and Oxidative Stress in AML12 Cells following Mild Hypothermia Treatment

The essential inflammation-associated proteins, including TLR4, iNOS, COX-2, TNF-α and HMGB1, were investigated by Western blotting (Figure 6A) to evaluate inflammation in AML12 cells after cold exposure. Western blot results showed the ratio of TLR4/β-actin, iNOS/β-actin, COX-2/β-actin, TNF-α/β-actin and HMGB1/β-actin increased distinctly in the MH treatment groups (Figure 6A–F). Furthermore, compared with control groups, the mRNA analysis of TLR4, TNF-α and HMGB1 also increased overtly (Figure 6G–I). We then examined the level of ROS and antioxidant proteins in mouse hepatocytes after MH treatment. ROS probe fluorescence staining result showed that with the increase of MH treatment time, the fluorescence increased, which means the ROS level increased (Figure 6J). Likewise, Nrf2, HO-1, NQO1, GCLC and GCLM proteins and mRNAs expressions visibly increased in the MH groups (Figure 6K–U).

### 3.6. The Effect of NLRP3/GSDMD Pyroptosis Pathway in the AML12 Cells following Mild Hypothermia Treatment

We investigated the effect of cold exposure of the pyroptosis pathway in AML12 cells by Western blot and qRT-PCR. These results indicated that the key proteins of the pyroptosis pathway, including NLRP3, ASC, Caspase-1, GSDMD, IL-1β and IL-18, were all activated following MH treatment (Figure 7A–L).

### 3.7. The Mechanism of Mild Hypothermia Treatment Activated Pyroptosis Signaling Pathway in AML12 Cells

It has been proved that cold exposure activates pyroptosis and antioxidant signaling pathways in in vivo and vitro experiments. However, it is still not clear what the relationship is between pyroptosis and antioxidant signaling pathways after MH treatment. Therefore, we used 100 µM NLRP3 inhibitor (MCC950) or 20 µM GSDMD inhibitor (necrosulfonamide) treatment AML12 cells for 3 h, then MH treatment AML12 cells for 6 h, and measured expression of the critical proteins involved in the pyroptosis and antioxidant pathway by Western blot. The results showed that MCC950 significantly inhibits the expression of NLRP3 and its downstream proteins, including Caspase-1, GSDMD and IL-1β (Figure 8A–E). Moreover, after inhibiting NLRP3 expression, the antioxidant-related proteins were also inhibited, such as Nrf2 and HO-1 (Figure 8A,F,G). MH treatment activated the expression of NLRP3, Caspase-1, GSDMD, IL-1β, Nrf2 and HO-1. However, these proteins were inhibited by adding MCC950 (Figure 8A–G). Then, the Western blot results showed that necrosulfonamide significantly inhibits the expression of GSDMD and its downstream proteins, such as IL-1β (Figure 8H–J), and the antioxidant-related proteins were also reduced, such as Nrf2 and HO-1 (Figure 8H,K,L). Although MH treatment induced GSDMD, IL-1β, Nrf2 and HO-1 expression, necrosulfonamide overtly reduced their expression (Figure 8H–L).

## 4. Discussion

Increasing studies have shown that cold exposure can result in liver damage [21,22]. In addition, the involvement of pyroptosis, inflammasomes and oxidative stress in the pro-inflammatory response is observed in many liver diseases, such as metabolic-associated fatty liver disease, nonalcoholic fatty liver disease and alcohol-associated liver disease [23,24]. However, very few studies have revealed pyroptosis’s underlying mechanisms of cold stress in liver damage. Therefore, the main purpose of our present study was to test the change of pyroptosis, inflammasomes and oxidative stress in the liver of mice following cold exposure, and the underlying mechanisms are shown in Figure 9. The work not only indicates that cold stress could induce oxidative stress and inflammatory damage in mouse liver by activating pyroptosis through the NLRP3/Caspase-1/GSDMD pathway, but also reveals that inhibition of the pyroptosis pathway had anti-cold stress leading to oxidative stress and inflammatory responses.

The chronic cold stress model was established based on our previous study [25]. First, it is well known that HSPs are a marker of stress in the organism and can resist specific stressful injuries [26]. The increased expression of HSPs indicates that the body’s adaptive and protective mechanisms are functioning, and also suggests that stress may have caused impaired cellular function [27,28]. Our results show that the protein expression of HSP70 and HSP90 increased evidently in the mice liver and AML12 cells following cold exposure, indicating that our chronic cold stress model instructed successfully and resulted in cold stress-induced liver injury of the mouse. HE staining is the most basic pathology staining technique. It is used to observe pathological changes of tissues and cells [29]. Results from our histopathological analysis demonstrated that the cells were swollen or even necrotic and disappeared, with no apparent intercellular demarcation in the cold group, and further proved cold stress-induced mouse liver pathological damages. It is well known that ALT, AST and LDH were indicators to evaluate liver function [30]. ALT is distributed in the cytoplasm, and the increase of ALT indicates that the membrane is damaged. AST is distributed in cytoplasm and mitochondria, and elevation suggests that the damage has reached organelle level [31,32]. If the liver is damaged or damaged, transaminases in liver cells enter the bloodstream, and elevated levels of ALT and AST in the blood signal liver disease. LDH is a metabolic enzyme, which is more distributed in the liver than in serum. Thus, when tissue is damaged, the enzyme is released into the blood and measured to aid in the diagnosis of liver disease [33]. In this study, the changes of AST, ALT and LDH in serum and liver of mice after chronic cold exposure were detected to determine the damage of liver function. Our results show that the dramatically elevated activity of ALT, AST and LDH in serum and liver was also confirmed that cold stress-induced liver injury.

Next, we found that cold stress can induce apoptosis in the liver and AML12 cells of mice according to Western blot test apoptotic marker protein levels results. In vivo, the Hoechst 33258 staining results from a morphological angle also confirmed that cold exposure induced apoptosis in mouse hepatocytes. Apoptosis is a type of cell death that depends on genetically coded signals or activity within dying cells. It is concerned with the growth and development of organs and tissues, immunity, metabolism and the removal of abnormal cells [34,35]. Furthermore, we observed that more cellular debris at the bottom of the medium, and “drawing” of the cells morphologically under the microscope after MH treatment, and the number of AML12 cells was reduced in a time-dependent manner after MH treatment. It is indicated that low temperature can inhibit the proliferation of liver cells and change the cell morphology to varying degrees, leading to the occurrence of cell damage. Based on our team’s previous findings that cold stress can result in neuroinflammation in mice, as a consequence, we measured the inflammatory mediator’s levels of TLR4, iNOS, COX-2, TNF-α and HMGB1 by Western blot and qRT-PCR [25]. The result met our expectations; above inflammatory mediator levels were all apparently up-regulated after cold exposure. In general, stress is always accompanied by oxidant stress [36]. Oxidant stress is an imbalance caused by excess free radicals and the body’s inability to counteract the damage they cause, which is a causative factor in various pathologies, such as cancer, diabetes, neurological diseases and liver disease [37,38,39,40]. In our present study, Western blot, qRT-PCR and immunohistochemistry analysis show that cold stress-induced Nrf2, HO-1, NQO1, GCLC and GCLM distinctly increased. In addition, the body’s antioxidant capacity as a defense ability is closely related to health degree, and antioxidant substances play an important role in protecting the damage caused by stress [41]. Among them, enzymes exist widely in eukaryotes and play an irreplaceable role in defending various tissues and organs against oxidative damage. For instance, GSH can scour free radicals and protect cell membrane and functional integrity; SOD protects the body by scavenging O2- and its level mainly reflects the body’s ability to scavenge oxygen free radicals; MDA mainly reflects the degree of lipid peroxidation in the body [42,43,44]. In this work, the activity of GSH and CAT notably down-regulated and the content of MDA clearly increased under cold exposure conditions. These results demonstrated that cold stress can seriously cause liver damage in mice by inducing apoptosis, inflammation and oxidant stress.

As far as we all know, pyroptosis is a form of cell death accompanied by an inflammatory response [45]. GSDMD, as a pivotal protein in the pyroptosis signaling pathway, can activate pyroptosis [46]. GSDMD is a soluble, inactive precursor protein consisting of GSDMD-N and GSDMD-C, two structural domains that can be separated by a bendable inter-domain linker [14]. Activated inflammatory cysteine aspartic proteases (Caspase-1, 4, and 5 in humans; Caspase-1 and 11 in mice) cleave GSDMD in structural interdomain linkers, resulting in the translocation of GSDMD-N into endoplasmic membrane leaflets and the subsequent pinching of two plasma membrane leaflets to aggregate into oligomers, resulting in large GSDMD pores that lead to cell permeability swelling and spontaneous cell death [47,48]. Previous studies have revealed that liver injury resulted in abundant damage and associated molecular patterns (DAMPs) being released, and further up-regulated and activated NLRP3 inflammasome, which led to pyroptosis [49]. Released cellular contents, such as IL-1β and IL-18 of inflammatory cytokine, can further aggravate pyroptosis [50]. In recent years, pyroptosis has become a hot topic of research. As the signaling pathways regulating mechanisms are gradually elucidated, researchers have also found that pyroptosis plays a considerable role in many human diseases and has important implications for disease treatment [51]. However, the molecular mechanism of regulating the expression of GSDMD in cold stress has not been elucidated. Thus, a possible regulator that targets GSDMD is of considerable importance. Our results show that cold exposure markedly activated NLRP3, ASC, Caspase-1, GSDMD, IL-1β and IL-18 in mice livers. Notably, growing evidence confirms that accumulation of ROS is a major pathway for NLRP3 inflammasome activation. For example, Wei Z et al. found cadmium (Cd)-induced ROS excess in duck renal tubular epithelial cells, which in turn led to cellular scorching in the ROS/NLRP3/Caspase-1 pathway [52]. In our current study, the reduction of NLRP3 could inhibit Caspase-1, GSDMD expression and alleviate serious inflammation induced by pyroptosis. The obtained results disclosed for the first time that NLRP3/Caspase-1/GSDMD axis-mediated hepatocyte pyroptosis in cold stress. Meanwhile, through using GSDMD inhibitor we showed a significant reduction in Nrf2 and HO-1 expression, which could be attributed to the down-released inflammation cytokines. The present study provided a novel mechanism and a solid foundation for further studies: that cold stress led to liver injury through activating the NLRP3/Caspase-1/GSDMD signaling pathway.

## 5. Conclusions

In conclusion, the results of this study indicate that cold stress can cause functional and pathological damage to mouse liver, inhibit the activity of AML12 cells, lead to changes in cell morphology, activate apoptosis, oxidative stress and pyroptosis, and promote the release of inflammatory cytokines, further aggravating liver injury. Meanwhile, we confirmed that cold stress activated pyroptosis and antioxidant signaling pathway through NLRP3/Caspase-1/GSDMD pathway. Inhibited pyroptosis, which further inhibited the antioxidant pathway and reduced inflammation cytokines’ release to alleviate liver damage. These findings clarified the mechanisms underlying the cold stress response and provided a basis for investigating new strategies to combat the effects of hypothermia.

## Figures and Tables

**Figure 1 biomolecules-12-00927-f001:**
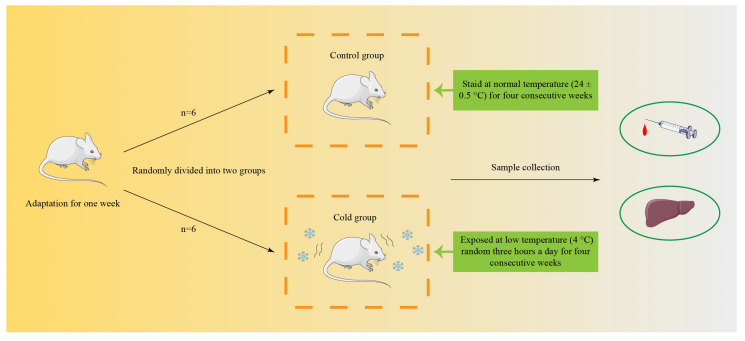
Timeline of the treatment protocol. The mice of control group were housed at a temperature of 22 ± 2 °C, and cold group was housed at 4 °C. Tissue samples were collected from each group on 3 weeks, *n* = 6 of each time point.

**Figure 2 biomolecules-12-00927-f002:**
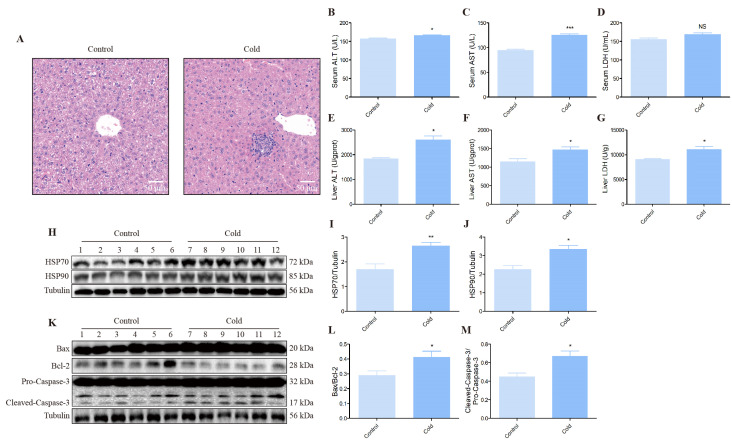
Effect of cold stress on the liver, activity of enzyme, level of stress and apoptosis-related protein. (**A**) The histopathological examination of liver of control and cold groups. (Scale bar = 50 µm). (**B**–**D**) The enzyme activity of ALT, AST and LDH in the serum of control and cold groups. (**E**–**G**) The enzyme activity of ALT, AST and LDH in the liver of control and cold groups. (**H**) HSP70, HSP90 and Tubulin protein levels in each group. (**I**,**J**) Quantification of expression levels based on expression ratios of HSP70/Tubulin and HSP90/Tubulin. (**K**) Bax, Bcl-2, Caspase-3 and Tubulin protein levels in each group. (**L**,**M**) Quantification of expression levels based on expression ratios of Bax/Bcl-2 and Cleaved-Caspase-3/Pro-Caspase-3. Data are presented as mean ± standard (*n* = 6). NS means not significant, * *p* < 0.05, ** *p* < 0.01, *** *p* < 0.001.

**Figure 3 biomolecules-12-00927-f003:**
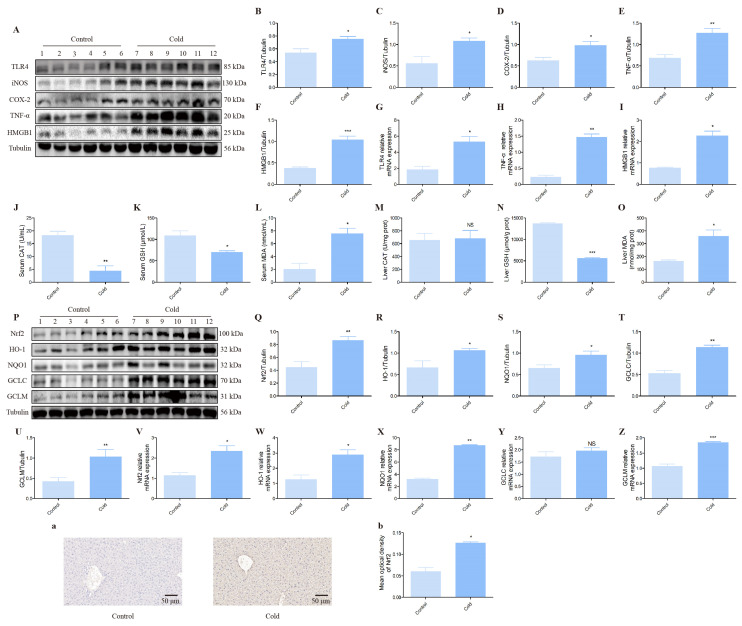
Effect of cold stress on inflammatory cytokines expression, oxidative stress-related indexes and protein levels assessed by Western blot and ELISA. (**A**) TLR4, iNOS, COX-2, TNF-α, HMGB1 and Tubulin protein levels in each group. (**B**–**F**) Quantification of expression levels based on expression ratios of TLR4/Tubulin, iNOS/Tubulin, COX-2/Tubulin, TNF-α/Tubulin and HMGB1/Tubulin. (**G**–**I**) The histograms of relative mRNA analysis of TLR4/β-actin, TNF-α/β-actin and HMGB1/β-actin. (**J**–**L**) The enzyme activity of CAT, GSH and content of MDA in the serum of control and cold groups. (**M**–**O**) The enzyme activity of CAT, GSH and content of MDA in the liver of control and cold groups. (**P**) Nrf2, HO-1, NQO1, GCLC, GCLM and Tubulin proteins levels in each group. (**Q**–**U**) Quantification of expression levels based on expression ratios of Nrf2/Tubulin, HO-1/Tubulin, NQO1/Tubulin, GCLC/Tubulin and GCLM/Tubulin. (**V**–**Z**) The histograms of relative mRNA analysis of Nrf2/β-actin, HO-1/β-actin, NQO1/β-actin, GCLC/β-actin and GCLM/β-actin. (**a**) Immunohistochemical scans of Nrf2. (Scale bar = 50 µm). (**b**) Average optical density values of Nrf2 proteins. Data are presented as mean ± standard (*n* = 6). NS means not significant, * *p* < 0.05, ** *p* < 0.01, *** *p* < 0.001.

**Figure 4 biomolecules-12-00927-f004:**
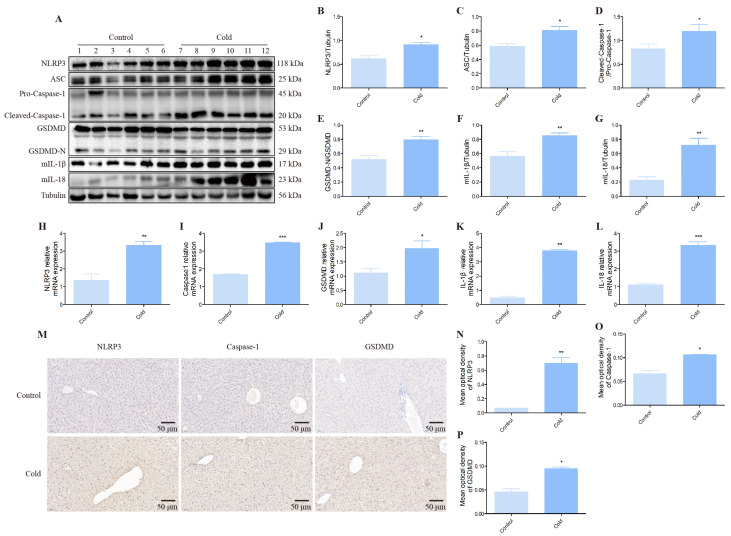
Effect of cold stress on NLRP3/GSDMD pyroptosis pathway assessed by Western blot, qRT-PCR and immunohistochemistry. (**A**) NLRP3, ASC, Caspase-1, GSDMD, IL-1β, IL-18 and Tubulin protein levels in each group. (**B**–**G**) Quantification of expression levels based on expression ratios of NLRP3/Tubulin, ASC/Tubulin, Cleaved-Caspase-1/Pro-Caspase-1, GSDMD-N/GSDMD, mIL-1β/Tubulin, mIL-18/Tubulin. (**H**–**L**) The histograms of relative mRNA analysis of NLRP3/β-actin, Caspase-1/β-actin, GSDMD/β-actin, IL-1β/β-actin, IL-18/β-actin. (**M**) Immunohistochemical scans of NLRP3, Caspase-1 and GSDMD. (Scale bar = 50 µm) (**N**–**P**) Average optical density values of NLRP3, Caspase-1 and GSDMD proteins. Data are presented as mean ± standard (*n* = 6). * *p* < 0.05, ** *p* < 0.01, *** *p* < 0.001.

**Figure 5 biomolecules-12-00927-f005:**
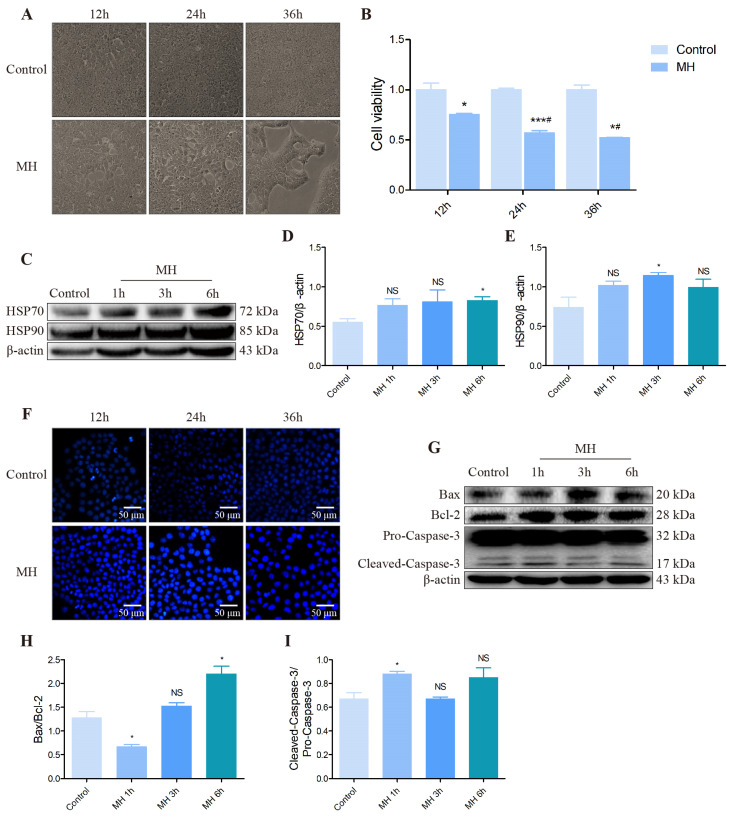
Effect of MH treatment on the cell morphology, number, HSPs and apoptosis-related protein expression in the AML12 cells. (**A**) The morphology of AML12 cells of control and cold groups. were observed under a microscope. (Scale bar = 50 µm). (**B**) MTT assay was used to detect the effect of MH on the growth and proliferation of AML12 cells. (**C**) The protein level of HSP70, HSP90 and β-actin in each group. (**D**,**E**) Quantification of expression levels based on expression ratios of HSP70/β-actin and HSP90/β-actin. (**F**) Hoechst 33258 staining was used to detect the effect of MH on AML12 cell apoptosis. (Scale bar = 50 µm). (**G**) The protein level of Bax, Bcl-2, Caspase-3 and β-actin in each group. (**H**,**I**) Quantification of expression levels based on expression ratios of Bax/Bcl-2, Cleaved-Caspase-3/Pro-Caspase-3. Data are presented as mean ± standard (*n* = 3). NS means not significant, * *p* < 0.05, *** *p* < 0.001, ^#^ *p* < 0.05. Where * represents comparison with the control group corresponding to their respective time, and ^#^ represents comparison with MH 12 h.

**Figure 6 biomolecules-12-00927-f006:**
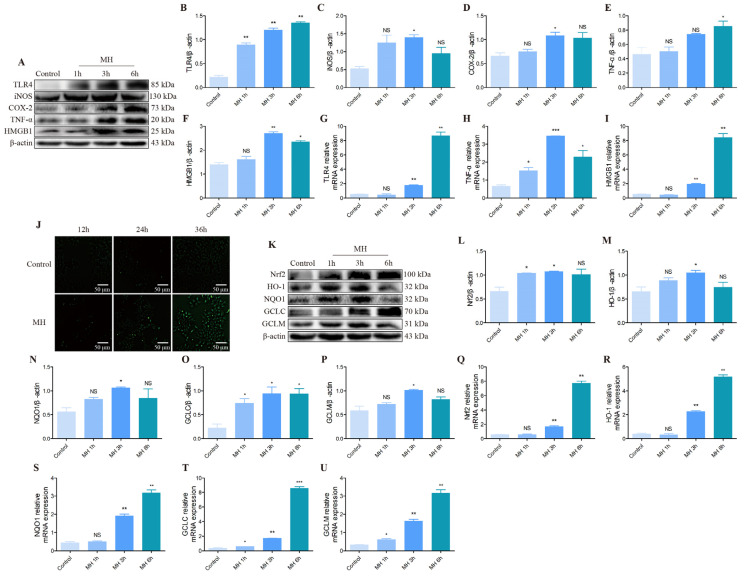
Effect of MH treatment on the expression of inflammatory cytokines and oxidative stress-related indexes in the AML12 cells. (**A**) TLR4, iNOS, COX-2, TNF-α, HMGB1 and β-actin protein levels in each group. (**B**–**F**) Quantification of expression levels based on expression ratios of TLR4/β-actin, iNOS/β-actin, COX-2/β-actin, TNF-α/β-actin and HMGB1/β-actin. (**G**–**I**) The histograms of relative mRNA analysis of TLR4/β-actin, TNF-α/β-actin and HMGB1/β-actin. (**J**) ROS probe fluorescence staining was used to detect the effect of MH on oxidative stress in AML12 cells. (Scale bar = 50 µm). (**K**) Nrf2, HO-1, NQO1, GCLC, GCLM and β-actin proteins levels in each group. (**L**–**P**) Quantification of expression levels based on expression ratios of Nrf2/β-actin, HO-1/β-actin, NQO1/β-actin, GCLC/β-actin and GCLM/β-actin. (**Q**–**U**) The histograms of relative mRNA analysis of Nrf2/β-actin, HO-1/β-actin, NQO1/β-actin, GCLC/β-actin and GCLM/β-actin. Data are presented as mean ± standard (*n* = 3). NS means not significant, * *p* < 0.05, ** *p* < 0.01, *** *p* < 0.001.

**Figure 7 biomolecules-12-00927-f007:**
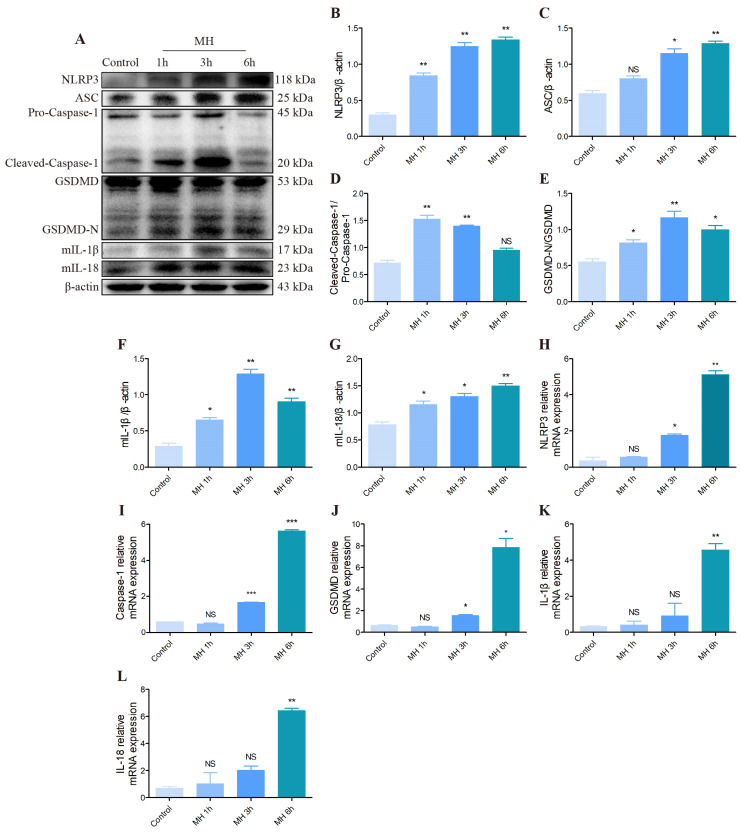
Effect of MH treatment on NLRP3/GSDMD pyroptosis pathway assessed by Western blot and qRT-PCR. (**A**) NLRP3, ASC, Caspase-1, GSDMD, IL-1β, IL-18 and β-actin protein levels in each group. (**B**–**G**) Quantification of expression levels based on expression ratios of NLRP3/β-actin, ASC/β-actin, Cleaved-Caspase-1/Pro-Caspase-1, GSDMD-N/GSDMD, mIL-1β/β-actin, mIL-18/β-actin. (**H**–**L**) The histograms of relative mRNA analysis of NLRP3/β-actin, Caspase-1/β-actin, GSDMD/β-actin, IL-1β/β-actin, IL-18/β-actin. Data are presented as mean ± standard (*n* = 3). NS means not significant, * *p* < 0.05, ** *p* < 0.01, *** *p* < 0.001.

**Figure 8 biomolecules-12-00927-f008:**
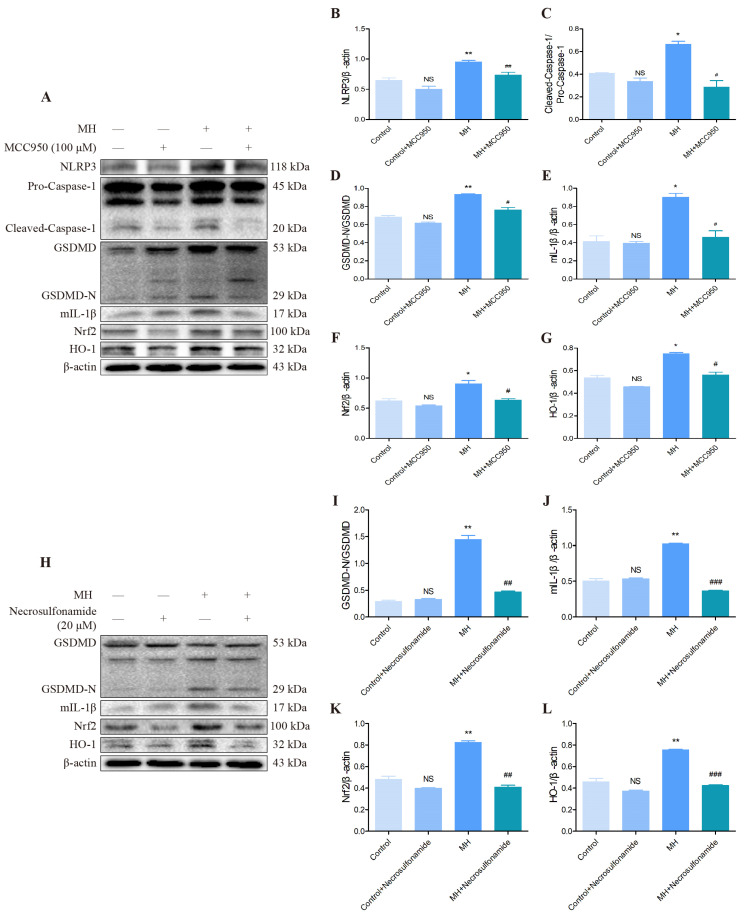
The relevant signaling pathway expression of pyroptosis of key proteins in AML12 cells after adding NLRP3 or GSDMD inhibitor treatment for 3 h as assessed by Western blotting and relative density analyses. (**A**) The protein level of NLRP3, Caspase-1, GSDMD, IL-1β, Nrf2, HO-1 and β-actin in each group after adding NLRP3 inhibitor (MCC950). (**B**–**G**) Analysis of the protein expression by the ratio of NLRP3/β-actin, Cleaved-Caspase-1/Pro-Caspase-1, GSDMD-N/GSDMD, mIL-1β/β-actin, Nrf2/β-actin, HO-1/β-actin. (**H**) The protein level of GSDMD, IL-1β, Nrf2, HO-1 and β-actin in each group after adding GSDMD inhibitor (necrosulfonamide). (**I**–**L**) Analysis of the protein expression by the ratio of GSDMD-N/GSDMD, mIL-1β/β-actin, Nrf2/β-actin and HO-1/β-actin. Data are presented as mean ± standard (*n* = 3). NS means not significant, * *p* < 0.05, ** *p* < 0.01, ^#^ *p* < 0.05, ^##^ *p* < 0.01, ^###^ *p* < 0.001. Where * represents comparison with the control group, and ^#^ represents comparison with the MH group.

**Figure 9 biomolecules-12-00927-f009:**
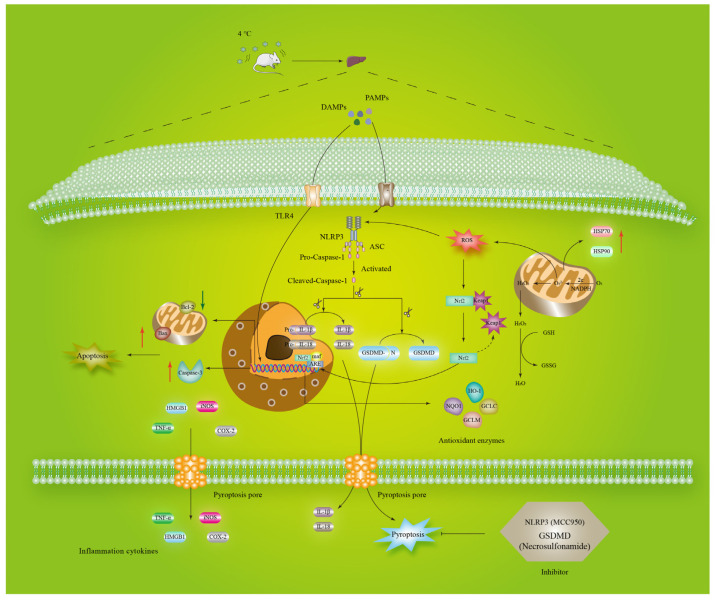
A proposed model for molecular mechanisms involving pyroptosis signaling pathway related to cold stress in the liver of mice.

## Data Availability

The datasets used during the current study are available from the corresponding authors on reasonable request.

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
