# Peer review of "Cold Stress Induced Liver Injury of Mice through Activated NLRP3/Caspase-1/GSDMD Pyroptosis Signaling Pathway"

_biomolecules, 2022, doi:10.3390/biom12070927_

Round 1
Reviewer 1 Report
I have gone through the manuscript. Topic is indeed interesting but needs extensive modifications. Authors should attempt to summarize how apoptosis and different mechanisms are interwoven together. Western blots for different mechanisms are not clear. For instance very confusing results for apoptosis. Pictures are not of high resolution. Authors should also analyze which pathway was involved in the induction of apoptosis? Was it through extrinsic pathway or intrinsic pathway. authors should analyze caspase-8 and caspae-9 for concrete evidence.
Oxidative stress and heat shock proteins have tight relation. What were the findings of researchers about heat shock proteins in Oxidative stress induced cells.
Why they did not analyze mitochondrial heat shock protein 90. It has important implications.

Author Response
Dear reviewers and editors,
We wish to thank you for the time and efforts you have spent reviewing our paper. Those comments are all valuable and very helpful for revising and improving our paper, as well as the important guiding significance to our research. We have studied these comments carefully and modified our manuscript accordingly. The changes have been highlighted in yellow throughout the manuscript. We hope the modifications have addressed all the shortcomings outlined. In particular, this revised manuscript of our resubmitted letter has been significantly improved as follows:
Reviewer 1
Question 1: I have gone through the manuscript. Topic is indeed interesting but needs extensive modifications. Authors should attempt to summarize how apoptosis and different mechanisms are interwoven together.
Answer: First of all, thank the expert for your affirmation of our work. And those comments are all valuable and very helpful for revising and improving our paper. Since we only tested apoptosis in this experiment, we did not analyze the relationship between apoptosis and other mechanisms too much. Please forgive me. In addition, we have seriously considered the teacher's suggestions and will include them in the follow-up study of the experiment. Thank you!
Question 2: Western blots for different mechanisms are not clear. For instance very confusing results for apoptosis. Pictures are not of high resolution.
Answer: Thanks for your nice suggestions and sorry for making you confused. Because the magazine requires only to upload manuscripts, the image definition may be compressed, the later will be uploaded original drawing according to the requirements of the magazine.
Question 3: Authors should also analyze which pathway was involved in the induction of apoptosis? Was it through extrinsic pathway or intrinsic pathway. authors should analyze caspase-8 and caspae-9 for concrete evidence.
Answer: Thank you for your professional advice. In this experiment, we detected apoptosis-related proteins including Bax, Bcl-2, Caspase-3, and the Hoechst 33258 staining was used to detect the apoptosis. In the follow-up study, we will analyze caspase-8 and caspase-9 level according to the teacher's suggestions.
Question 4: Oxidative stress and heat shock proteins have tight relation. What were the findings of researchers about heat shock proteins in Oxidative stress induced cells.
Answer: Thank you for your valuable advice. In this experiment, we found that heat shock proteins as stress marker proteins, the protein expression of HSP70 and HSP90 increased evidently in the mice liver and AML12 cells following cold exposure. But we did not take into account oxidative stress on heat shock proteins, this is a lack of thoughtfulness on our part. Thank you for pointing out the problem. We will consider and verify the relationship between them in the follow-up experiment.
Question 5: Why they did not analyze mitochondrial heat shock protein 90. It has important implications.
Answer: Thank you for pointing this out. After listening to the teacher's advice, we looked up the relevant materials and thought it was a very good idea. We will add the teacher's suggestions in the follow-up in-depth research.

Reviewer 2 Report
The manuscript entitled “Cold stress induced liver injury of mice through activated NLRP3/Caspase‐1/GSDMD pyroptosis signaling pathway” aimed to investigate the effects of chronic cold exposure on liver function, apoptosis, oxidative stress and inflammation in mice by establishing a mouse model of chronic cold exposure, and to further investigate whether pyroptosis pathways are involved in the process of chronic cold exposure. Authors showed that inflammatory cell infiltration and other pathological changes in liver cells and the activity of liver enzyme increased in serum and liver of cold‐exposed mice, indicating that cold stress may result in liver injury. Overall, they concluded that their findings provide a new understanding of the underlying mechanisms of the cold stress response, which can inform the development of new strategies to combat the effects of hypothermia.
In 2.12 ROS assay, please explain in more details how ROS production was analysed after the photos were taken. Didi authors use an image analysis system. Were quantitative data of fluorescence intensity obtained from regions of interest? Please specify.
In Conclusion section, please rephrase the sentence to make it more clearer “In the present study, the results were used to characterize cold stress can impair liver function and pathological damage to the liver in mice, inhibit AML12 cells activity, promote cell morphology altered, induce apoptosis, oxidant stress and pyroptosis and release inflammation cytokines”
Minor remarks:
Place space between reference number and text throughout the paper (e.g. Introduction, references 5, 6, 7 etc.)
Use min instead of minutes and h instead of hours to be the same throughout the paper
Page 2, line 88 - Its either 20.00 hours or 8.00 PM, using both is redundant
Author Response
Dear reviewers and editors,
We wish to thank you for the time and efforts you have spent reviewing our paper. Those comments are all valuable and very helpful for revising and improving our paper, as well as the important guiding significance to our research. We have studied these comments carefully and modified our manuscript accordingly. The changes have been highlighted in yellow throughout the manuscript. We hope the modifications have addressed all the shortcomings outlined. In particular, this revised manuscript of our resubmitted letter has been significantly improved as follows:
Reviewer 2
Question 1: The manuscript entitled “Cold stress induced liver injury of mice through activated NLRP3/Caspase‐1/GSDMD pyroptosis signaling pathway” aimed to investigate the effects of chronic cold exposure on liver function, apoptosis, oxidative stress and inflammation in mice by establishing a mouse model of chronic cold exposure, and to further investigate whether pyroptosis pathways are involved in the process of chronic cold exposure. Authors showed that inflammatory cell infiltration and other pathological changes in liver cells and the activity of liver enzyme increased in serum and liver of cold‐exposed mice, indicating that cold stress may result in liver injury. Overall, they concluded that their findings provide a new understanding of the underlying mechanisms of the cold stress response, which can inform the development of new strategies to combat the effects of hypothermia.
Answer: Thank you for your comments and I hope to learn more from you. And thank you very much for your recognition of our work.
Question 2: In 2.12 ROS assay, please explain in more details how ROS production was analysed after the photos were taken.
Answer: Thank you for your valuable suggestion. Here, ROS were simply observed under a fluorescence microscope, therefore, we did not analyze ROS production.
Question 3: Didi authors use an image analysis system. Were quantitative data of fluorescence intensity obtained from regions of interest? Please specify.
Answer: Thank you for your advice and sorry for making you confused. In Figure 6 J, these photos were photographed by fluorescence microscope, and we repeated many experiments to reduce the results due to objective reasons.
Question 4: In Conclusion section, please rephrase the sentence to make it more clearer “In the present study, the results were used to characterize cold stress can impair liver function and pathological damage to the liver in mice, inhibit AML12 cells activity, promote cell morphology altered, induce apoptosis, oxidant stress and pyroptosis and release inflammation cytokines”.
Answer: Thank you for your valuable suggestions. We have revised it to “In conclusion, the results of this study indicate that cold stress can cause functional and pathological damage to mouse liver, inhibit the activity of AML12 cells, lead to changes in cell morphology, activate apoptosis, oxidative stress and pyroptosis, and promote the release of inflammatory cytokines, further aggravating liver injury” according to your advice.
Minor remarks:
Question 5: Place space between reference number and text throughout the paper (e.g. Introduction, references 5, 6, 7 etc.)?
Answer: Thank you for pointing this out. We have revised this mistake according to your advice. And the full text is checked and revised.
Question 6: Use min instead of minutes and h instead of hours to be the same throughout the paper?
Answer: Thank you for your valuable suggestions. We have revised min instead of minutes and h instead of hours according to your advice.
Question 7: Page 2, line 88 - Its either 20.00 hours or 8.00 PM, using both is redundant?
Answer: Thank you for your correction. We have revised this mistake according to your advice.

Round 2
Reviewer 1 Report
Looks in good form now.